# Multi-Terminal Nonwoven Stochastic Memristive Devices Based on Polyamide-6 and Polyaniline for Neuromorphic Computing

**DOI:** 10.3390/biomimetics8020189

**Published:** 2023-05-03

**Authors:** Nikita Prudnikov, Sergey Malakhov, Vsevolod Kulagin, Andrey Emelyanov, Sergey Chvalun, Vyacheslav Demin, Victor Erokhin

**Affiliations:** 1National Research Centre “Kurchatov Institute”, 123182 Moscow, Russia; 2Moscow Institute of Physics and Technology (National Research University), 141701 Dolgoprudny, Moscow Region, Russia; 3Institute of Materials for Electronics and Magnetism, Consiglio Nazionale delle Ricerche (IMEM-CNR), 43124 Parma, Italy

**Keywords:** nonwoven material, organic memristive device, reservoir computing, stochastic device, multi-terminal memristive device

## Abstract

Reservoir computing systems are promising for application in bio-inspired neuromorphic networks as they allow the considerable reduction of training energy and time costs as well as an overall system complexity. Conductive three-dimensional structures with the ability of reversible resistive switching are intensively developed to be applied in such systems. Nonwoven conductive materials, due to their stochasticity, flexibility and possibility of large-scale production, seem promising for this task. In this work, fabrication of a conductive 3D material by polyaniline synthesis on a polyamide-6 nonwoven matrix was shown. An organic stochastic device with a prospective to be used in reservoir computing systems with multiple inputs was created based on this material. The device demonstrates different responses (output current) when different combinations of voltage pulses are applied to the inputs. The approach is tested in handwritten digit image classification task in simulation with the overall accuracy exceeding 96%. This approach is beneficial for processing multiple data flows within a single reservoir device.

## 1. Introduction

Recently, neuromorphic systems are widely considered as a paradigm for brain-like information processing. It is expected that such systems will allow to reach several advantages, such as parallel information processing, low energy consumption, capability of unsupervised learning, etc. According to [1], the system must have at least five important properties to be considered as a neuromorphic one. We do not repeat them here because at the present state-of-the-art level, it is considered that mimicking at least one (or, better, more) essential requirements would provide an important step towards the realization of a system capable of neuromorphic information processing. Probably, the main feature of biological information processing systems (nervous system in general and brain in particular) is the fact that memorizing and processing the information is performed by the same elements (neurons, where synapses (connections of neurons) play a very important role, varying the connection weights of neuronal cells according to special rules, as a Hebbian one). It is to note that this consideration is also used for the realization of artificial neural networks: elements connections (synapse analogs) of threshold nodes (neurons) can vary their weight function according to the performed training procedure. However, even in the case of a rather simple network, containing more than two neuron layers, the training procedure is rather complicated: it is necessary to know not only the current state of values of weight matrices of each layer of synapse mimicking elements but also their temporal derivatives during iterations. Of course, it is not the case in brain-mimicking systems. The brain is a very complicated system, composed of 10^11^ neurons and 10^15^ synapses [2]. Learning the system at the level of single pairs of neurons is rather well explained using the current modification of Hebbian Rule [3], known as Spike-Timing-Dependent Plasticity (STDP) [4]. Currently, STDP is considered as a main algorithm of unsupervised learning, establishing a causal relationship of two events (spiking of pre- and post-synaptic neurons) according to the time delay and consequence of them. It is to note that the main element of the presented study (polyaniline-based memristive device) was successfully used both for the realization of artificial systems, allowing STDP-like learning [5], and for the connection of two live neurons from rat cortex [6]. In the last work, when the synaptic connection was established, the delay of the pre- and post-synaptic spikes of neurons was about 3 ms, corresponding well to the situation in brains. This work can be considered as a direct demonstration that a polyaniline memristive device can be really considered as an electronic analog of biological synapses.

However, in the case of the system as a whole (brain), it is not possible to apply these rules. In reality, we deal with a system with stochastic organization of elements even with distant connections which, however, can be structured during learning. Learning occurs through feedback, adjusting all possible weight function values of the intermediate medium elements for the realization of appropriate outputs (e.g., motor neurons) corresponding to vectors of inputs (signals from sensor systems). Thus, it is not necessary to know weight function values of all intermediate synapses. It is only necessary that the intermediate medium has nonlinear properties and its elements have short-term volatile memory properties. Learning in this case is performed just by the correction of the weight function values of synapse-like elements, connecting the nonlinear medium with sensory elements, on the one hand, and executive elements, on the other hand. This requirement corresponds well to the concept of reservoir computing approach.

A reservoir computing system is a machine learning framework for dynamical processing of spatial and temporal series of data [7]. Such systems usually consist of two main parts. The first part is a reservoir itself, which takes a sequence of time pulses as an input and changes its state depending on the temporal and shape distribution of the pulses due to short-term memory effects. The output values of the reservoir, which are current amplitudes transformed from input voltage reading pulses applied to the system just after the main sequence, are passed to the second part of the system–readout layers. The training procedure consists in proper selection of readout layer weights that provide the best correlation between the target output and input transformed by the reservoir state. The main advantage of such systems over traditional neural networks is reduced time and energy costs, because the learning process occurs only in readout layers, while the reservoir part is not trainable.

Originally, reservoir computing was implemented in software, but recently hardware realizations based on different physical phenomena have become very common [8]. One of the most investigated approaches is electronic reservoir based on volatile memristive devices. The latter change their resistance in response to applied voltage and gradually return to the initial state under zero bias in a short period of time [9]. Due to the extremely wide range of dynamical characteristics provided by different memristive devices, these systems have found applications in various tasks from image classification [10,11] to processing of biological signals, behaving as an internal part of biosensors [12].

Different approaches can be used to implement a hardware memristive reservoir, but the most straightforward one is to use a strictly deterministic architecture: 1 input–1 memristor–1 output [13,14]. However, recent works have demonstrated stochastic reservoirs based on organic materials with a large number of parallel inputs and outputs [15,16]. This approach allows to significantly decrease the number of devices used and, thus, the dimension of the input data, which leads to a reduction in the complexity and power consumption of the entire network. Three-dimensional memristive devices based on nonwoven materials, which provide native stochasticity, flexibility and possibility of large-scale production [17], seem promising for organic-based reservoir systems. Moreover, such self-organized and naturally highly interconnected architecture could be considered as a cheap but effective alternative for passive crossbar architectures [18].

Organic memristive devices can be produced with the use of different materials such as polyaniline (PANI) [5], oligo- [19] and polythiophenes [20,21] and small molecules [22]. However, due to the simplicity of synthesis, stability and higher conductivity, PANI proved to be the most suitable for production of nonwoven conductive materials. In fact, the possibility of realizing stochastic structures by vacuum treatment of polyethylene oxide with successive PANI deposition has been shown [23], self-assembling of specially synthesized block-copolymers, containing insulating and solid electrolyte parts [24], and by electrospinning [25]. The latter structures could be fabricated in three different ways: by simultaneous electroforming of PANI and a fiber-forming polymer in a mixture [26,27,28], coaxial formation of fibers [29], and PANI synthesis on electrospun non-conductive polymer matrix [30,31,32,33]. The third method brings the most flexibility in choosing matrix polymer according to the desired properties of the final structure.

In this work, we demonstrate nonwoven hybrid materials based on conductive polyaniline synthesized on a non-conductive polyamide-6 fibrillar matrix. We also present a concept of using a nonwoven stochastic memristive device with multiple inputs and single output, which has a potential for application in reservoir computing systems. The approach is validated by a simulation of reservoir-like computing system using experimental current values distributions for eight states of the presented device. The system demonstrates more than 96% accuracy rate, while reducing the number of reservoir units, as well as readout network complexity as much as triple, yet providing the separation of different input combinations of time sequences. These results are promising and suggest that the presented device can perform image classification tasks with high accuracy and efficiency.

## 2. Materials and Methods

For fabrication of nonwoven materials we used polyamide-6 (PA) (Volgamid 27, Kuibyshevazot, Togliatti, Russia) with a relative viscosity of 2.7, aniline hydrochloride (99%, Across Organics, Geel, Belgium), ammonium peroxodisulfate (98+%, Across Organics), 1,1,1,3,3,3-hexafluoroisopropanol (HFIP) (99%, PiM-Invest, Moscow, Russia), hydrochloric acid (Sigma Tech, Moscow, Russia). All reagents were used without any additional purification.

To prepare a spinning solution with a concentration of 6%, the required PA and HFIP amounts were mixed with a magnetic stirrer for 24 h. A custom laboratory device was used for electrospinning. The solution was poured into a plastic syringe with a volume of 20 mL, which was placed in a syringe pump (DSH-08, “Visma-Planar”, Minsk, Belarus). The polymer flow rate was set to 3 mL h^−1^. A rotating cylinder collector with a diameter of 15 cm was placed at a distance of 30 cm from the syringe. The applied voltage (25 kV) was created by a high-voltage source Spellman SL130PN30. At the end of the electrospinning process, the nonwoven fabricated system was removed from the collector and kept under vacuum to remove the residual solvent.

To obtain a polyaniline layer, nonwoven material fragments of size 3 × 1.5 cm were formed, fixed in glass clips, after which the synthesis of PANI was carried out according to the IUPAC method [34] with a 0.1 M concentration of reagents and a synthesis time of 2 h. Then, the excess PANI were washed off from the samples, and the latter were dried for 24 h.

FTIR spectra of the samples were obtained using a Nicolet iS5 (Thermo Fisher Scientific, Waltham, MA, USA) spectrometer with an iD5 ATR accessory. The spectra were recorded in the region of 4000–550 cm^−1^. SEM imaging was performed using a Phenom XL instrument (Thermo Fisher Scientific, USA) equipped with backscattered electron detector. Pressure was set to 10 Pa, and the accelerating voltage was 5 kV. Image processing and evaluation of fiber diameters were carried out by the ImageJ 1.49v program. The study of surface properties was carried out by determining the water drop contact angles using the KRUSS DSA30E system with the 5 µL water test droplets.

A glass plate with copper tape electrodes was used as a supporting substrate for the memristive device. A silver wire with a diameter of 125 µm was mounted on a plate in the center between electrodes and covered with a layer of gel-electrolyte—an essential part of the device, providing a medium for redox reactions and responsible for the resistance switching. To prepare the latter, we made an aqueous solution of polyethylene oxide (Sigma-Aldrich, Burlington, MA, USA, Mw = 5,000,000 g/mol) with a concentration of 30 g L^−1^ in water with the addition of lithium perchlorate (Sigma-Aldrich, concentration 0.19 M) and hydrochloric acid (concentration 0.1 M). Then, a fragment of nonwoven material, which was formed according to a special template, was put on the electrolyte drop and fixed on the substrate by one more layer of copper tape. Another layer of electrolyte was deposited on the top of the structure in order to provide a contact of the silver wire and bulk of the nonwoven material. After all the operations, the device was dried at room temperature for 24 h. The switching kinetics was measured using a Keysight B2902A Precision Source/Measure Unit (SMU). The reservoir experiment was carried out using an NI PXIe 4140 SMU.

A formal neural network was used to recognize digits from the multi-terminal stochastic device response. The network was composed of 252 inputs, 100 neurons in the hidden layer and 10 outputs. ReLU and Softmax were used as activation functions for neurons in the hidden and output layers, respectively. The Adam optimizer, which is based on adaptive estimation of first-order and second-order moments, was used. This setup is suitable for datasets with thousands of training samples or more in terms of both training time and validation score. The learning rate was constant at 10^−3^, and cross-entropy was used as a loss function. Inputs of the network were generated randomly based on the experimental distributions for the multi-terminal stochastic memristive device. The simulation was performed with scikit-learn framework.

## 3. Results

### 3.1. Material Characterization

The first preparation stage was connected to the formation of fibrillar network, using a well-known material suitable for the application of electrospinning. The obtained nonwoven structure made of polyamide-6 (Figure 1a) is characterized by cylinder-shaped fibers and smooth surface morphology with an average diameter of ~700 nm, which is typical for electrospinning of this polymer from a solution [35,36,37]. After the synthesis of PANI on this template (Figure 1b), the average diameter of the fibers was increased by about 10%, i.e., the thickness of the conductive layer of PANI on polyamide fibers does not exceed 50 nm. It is also worthy of mention that the color of the matrix, composed from the source material (pure polyamide) was white, while at the end of synthesis of PANI, it acquired a dark green color, characteristic for the presence of PANI. It is also noteworthy that the polymerization took place in the entire volume of the material, which is visible on its cross sections.

In the IR spectra (Figure 2) of the initial fibers from pure PA, there are characteristic bands of 3295 cm^−1^ (stretching vibrations of N–H bonds), 1640 cm^−1^ (amide I, stretching vibration C=O), 1547 cm^−1^ and its overtone at 3084 cm^−1^ (amide II, bending vibrations N–H and stretching C–N), 1205–1279 cm^−1^ (amide III, bending vibrations N–H + bending C=O + stretching C–C) [37,38]. New absorption bands appear on the spectra of materials after the synthesis of PANI (including partially overlapping with PA bands) in the region of 800–830 cm^−1^ (bending vibrations of 1,4-disubstituted benzene derivatives), 1304 cm^−1^ (C–N aromatic amines), 1612 and 1490 cm^−1^ (vibrations of C–C in benzenoid and quinoid rings), 3050–3100 cm^−1^ (stretching vibrations of C-H in aromatic rings) and 3244 cm^−1^ (stretching vibrations of N–H bonds) [25,39].

The initial PA nonwoven material is hydrophobic; however, during the synthesis of PANI in an aqueous medium, the material loses its hydrophobicity [34], which allows the synthesis of PANI on fibers throughout the whole thickness of the material. Hybrid electrically conductive material PA-PANI demonstrates rapid absorption of applied water droplets, which makes it suitable for the assembly of memristive devices (because impregnation is required throughout the thickness of the sample with a water-based electrolyte, which cannot be achieved for a hydrophobic material).

### 3.2. Multi-Terminal Device

To test the possibility of using of the realized system, based on PA-PANI nonwoven material for reservoir computing tasks, the stochastic memristive device structure with three inputs and a single output terminal was fabricated according to the scheme shown in Figure 3a. A fragment of nonwoven material was mounted on a glass plate with copper tape electrodes and with attached silver wire, serving as working and reference electrode. The image of the assembled device is shown in Figure 3b, while the connection diagram and operation scheme in the reservoir regime mode are shown in Figure 3c. Every pixel of the image row is supposed to be fed to a separate input of the device in the form of a voltage pulse of a certain amplitude. The value of the current from the common output of the device, which is measured during the reading phase right after the coding pulse sequence, is supposed to be used as a response signal.

Switching of PANI memristive devices is caused by oxidation and reduction processes of PANI film, which leads to an increase and decrease of its conductivity, respectively. Oxidation and reduction processes occur under different voltage bias values (>0.5 V and <0.2 V, respectively), and the device is quite stable within the range between these values, which makes it possible to read its current resistance without influencing it.

Figure 4a shows change in conductivity of the device by application of alternating voltage pulses with an amplitude of 0.8 V (potentiation) and −0.3 V (depression) to each of the inputs (only one at a time) with a duration of 60 s. At the same time, reading voltage with an amplitude of 0.4 V was applied to the other two inputs. The chart shows the response current value measured at the common output terminal. Although the device demonstrates slow response and relatively weak voltage-induced change in conductivity, the final values of the current response for different combinations of input voltages can be clearly distinguished. Therefore, such devices could be potentially used in reservoir computing tasks for classification of different patterns in multi-terminal mode. The observed timescale limits the number of possible classification tasks. However, the switching speed of the devices can be reduced by miniaturization of the active switching region like it was previously shown for the deterministic PANI-based memristive devices [40].

The results of the single stochastic memristive device operation in a multi-terminal mode are shown in Figure 4b. All eight possible patterns consisting of different combinations of three bits were mixed in a random order, and the resulting set was applied to the system for three times in a row. Pixel “0” was represented by 0.4 V voltage, while pixel “1” was represented by 0.8 V. The response output current was measured during the entire 60 s period of voltage application with 0.5 s resolution. We repeated the experiment three times to exclude the random factors. One can see from the figure that the current values almost reach stationary states after 30 s of voltage exposure, while being well separable from each other. Despite the fact that significant overlapping could be noted for some states (“110”, “001” and “100”), the impact of this fact on the overall system error in recognition tasks can be significantly reduced by processing in the readout layers of the system following the reservoir.

### 3.3. Neural Network Simulation

The potential of the memristive device presented in this study for real-world applications is immense. In order to demonstrate its capabilities, we performed a simulation of image classification of handwritten digits from the binarized MNIST (Modified National Institute of Standards and Technology) database. As depicted in Figure 3c, every pixel of the image is converted to a voltage pulse that is then applied to the memristive device. Our approach enables the device to process three pixels at a time, thereby reducing the number of network inputs by three times. To achieve this, the images were reduced to the size of 28 × 27 pixels, which were then divided into sections consisting of 3 pixels each, as the device has 3 input terminals. In future hardware implementations, each group of pixels is to be applied to an individual stochastic memristive device.

For the simulation, we selected 3 different distributions of states of the single device from Figure 4b, which were taken after 10, 30 and 60 s of continuous potentiation. After 10 s, the device did not reach the saturation current values, and some states were significantly overlapped. However, after 30 s, the current almost finished increasing, and there was complete saturation with good separation after 60 s. This demonstrates that the memristive device has the potential to achieve high accuracy in image classification tasks, and its processing speed can be optimized by adjusting the time of potentiation.

The simulation results are presented in Figure 5 using confusion matrix, which clearly show that the percentage of error is as low as 1% for the worst cases of individual digits. The overall accuracy of the system after averaging over 10 runs for each case increases 96% for all 3 distributions. These results are very promising and demonstrate the potential for the presented device to perform image classification tasks with high accuracy and efficiency. Additionally, it was observed that the approach remained robust even when there was overlapping of some states, which was due to the efficient separation provided by the subsequent readout layers.

## 4. Discussion

In this paper, we have proposed an organic stochastic memristive device system based on conductive nonwoven materials made of polyamide-6 and polyaniline with multiple terminals, which can be used as inputs or outputs. We have also presented a concept of using such a system for reservoir computing tasks to process temporal data from several sources simultaneously within a single device. The presented approach helps to reduce the number of network inputs, which results in faster processing times. Furthermore, it has the potential to achieve high accuracy with reduced memory requirements. These advantages mark the memristive device as a promising candidate for a wide range of applications, including image and speech recognition, data analytics and machine learning. Although the device demonstrates good differentiation of states depending on the input combination, slight change in conductivity is a constraint for using such devices for processing of time sequences of data. However, the time required to process a single voltage pulse should be significantly reduced to make it possible to process an entire sequence of pulses at each of the inputs. This could be potentially achieved by miniaturization of the device, which was previously proved on thin-film PANI memristive devices. From our point of view, the active area of the device, which is the area between two planar electrodes, could be scaled down to several hundreds of µm^2^ without any physical limitations. In one of the previous works, the authors have demonstrated a working device based on a single fiber only [11]. However, single fiber is not enough to maintain the stochastic nature of the network since the diameter of the former is about 500 nm. Additionally, miniaturization requires a more complex fabrication process, such as using a micromanipulator, since the fragility of nonwoven materials complicates the process fabrication process and stability of the whole system. It seems important that the stochastic system must allow multiple signal pathways. In this case, the network will be always in a dynamic equilibrium state, and small damages to the structure as well as presence of external noise will not significantly affect its capability of objects identification. These problems are priority for future studies.

## 5. Conclusions

The present study is a step towards using systems with stochastic structure for neuromorphic applications. In fact, previously reported works based on the stochastically organized fibrillar systems [23,41], fabricated by vacuum treatment, have demonstrated the possibility of rather simple training tasks (reinforcement and inhibition of conductivity between selected pairs of attached electrodes). Moreover, the free-standing nature of the formed systems resulted in the fast degradation of learning capabilities even when supporting porous “skeletons” templates were used [41] (however, the stability in the latter case was slightly improved). Thus, these works [23,41] have demonstrated a principial possibility of the realization of PANI-based stochastic systems with learning capabilities. However, the mentioned studies revealed an important problem: very low stability of electrical properties in such systems.

As the next step, a 3D system, composed from PEO-PANI fiber and fabricated by electrospinning method [17], was realized and studied. However, relatively low conductivity of the realized structures (about 10^−3^ S/cm) was a key parameter, limiting the applicability of such systems for neuromorphic applications. A significant result was obtained, when the stochastic 3D system was realized using specially synthetized block copolymers, allowing self-assembling due to the phase separation [24]. This work has demonstrated very interesting results, including the possibility of short- or long-term potentiation or inhibition of signal pathways according to the applied training algorithm. However, only two conductivity states were considered within this work: low and high ones. This is a serious limitation for exploiting this system both for neuromorphic and artificial neural networks applications.

In the current work, we have described the system that eliminates all the drawbacks of the previously described approaches. In fact, it is significantly more stable than fibrillar systems fabricated by vacuum treatment [23,41]; its conductivity is much higher with respect to 3D systems of PEO-PANI fibers, fabricated by electrospinning method [17]; it has more distinct resistive states compared to samples fabricated by self-assembling of specially synthetized block-copolymers [24]. Summarizing, it seems that the reported system is very prospective for both neuromorphic and artificial neural networks applications, because it has 3D stochastic organization with integrated information memorizing and processing properties, allowing training with multiple distinct resistive states.

## Figures and Tables

**Figure 1 biomimetics-08-00189-f001:**
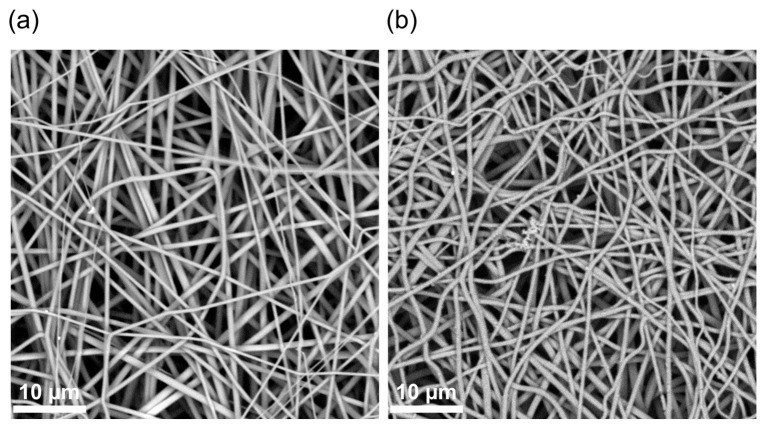
SEM images of nonwoven material of pure polyamide-6 (**a**) and covered by polyaniline after synthesis (**b**).

**Figure 2 biomimetics-08-00189-f002:**
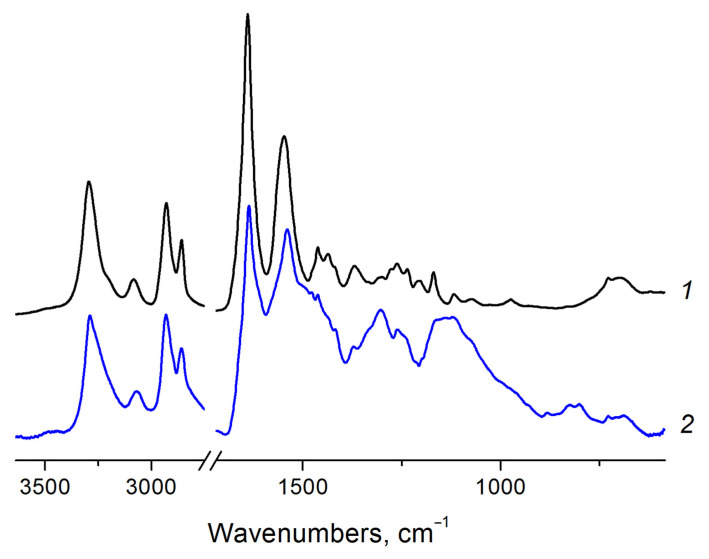
FTIR spectra of nonwoven material of pure polyamide-6 (1) and covered by polyaniline after synthesis (2).

**Figure 3 biomimetics-08-00189-f003:**
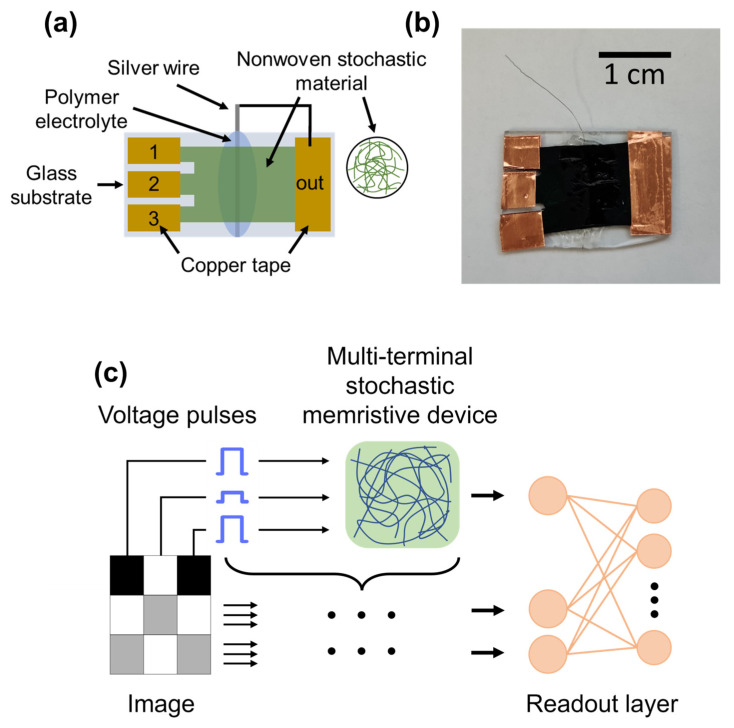
(**a**) Assembling diagram of the organic stochastic multi-terminal memristive device. (**b**) Image of the fabricated device. (**c**) Concept of operation of organic stochastic multi-terminal memristive device in a reservoir computing system.

**Figure 4 biomimetics-08-00189-f004:**
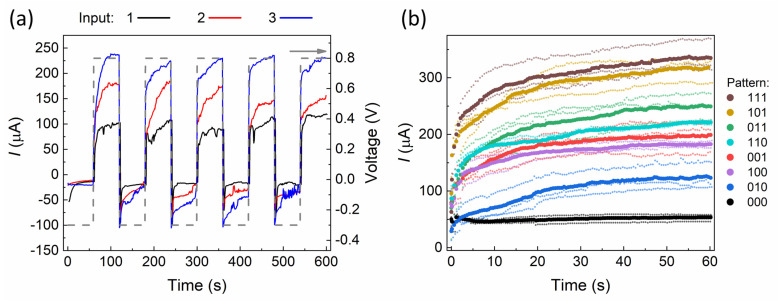
(**a**) Output current change induced by application of high voltage to different inputs of the device (dash-line represents the voltage pulses). (**b**) The results of the reservoir experiment of applying different input voltages combinations to the inputs of the device with 3 experimental semitransparent curves and highlighted average for each pattern.

**Figure 5 biomimetics-08-00189-f005:**
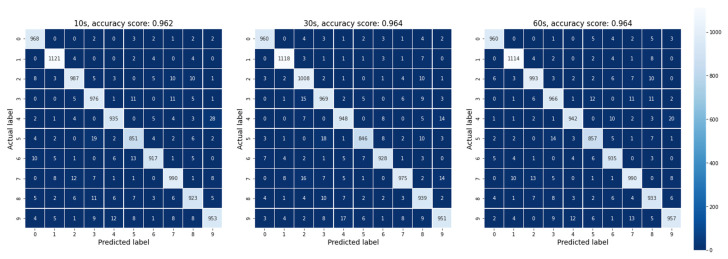
Confusion matrices summarizing the simulation results of handwritten digit recognition task for 3 different distributions of the device states after 10, 30 and 60 s of continuous potentiation.

## Data Availability

The data supporting the findings of this study are available from the corresponding author upon reasonable request.

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
