# Peer review of "Multi-Terminal Nonwoven Stochastic Memristive Devices Based on Polyamide-6 and Polyaniline for Neuromorphic Computing"

_biomimetics, 2023, doi:10.3390/biomimetics8020189_

Round 1

Reviewer 1 Report

In their work entitled "Nonwoven stochastic memristive devices based on polyamide-6 2 and polyaniline for reservoir computing", the authors are propose a synthetic material that could be used as for the hardware implementation of a reservoir computer.

I find the idea interesting, and the work appears to be sound. However, my main concern is the claim that this device could perform reservoir computing. While plausible, this claim has not be proven in this work. The authors have just shown that they have developed the capability to fabricate a nonwoven stochastic memristive web.

I would therefore recommend publication if only if "reservoir computing" is removed from the title and toned down as claim in the paper (it is OK to state that this device could be used for reservoir computing, though).

Author Response

Reviewer 1

I find the idea interesting, and the work appears to be sound. However, my main concern is the claim that this device could perform reservoir computing. While plausible, this claim has not be proven in this work. The authors have just shown that they have developed the capability to fabricate a nonwoven stochastic memristive web.

I would therefore recommend publication if only if "reservoir computing" is removed from the title and toned down as claim in the paper (it is OK to state that this device could be used for reservoir computing, though).

The title of the article was changed to avoid misleading the readers. We have also made some corrections in the text to emphasize the potentiality of using such devices in reservoir computing systems instead of demonstrating state-of-the-art technology (New section 3.3). We have also modified the Introduction section and added Conclusions section.

Reviewer 2 Report

In this work, authors reported on the fabrication of nonwoven stochastic memristive devices for reservoir computing. Despite the topic is interesting for a wide audience, several improvements should be performed before publication.

1) The title could be misleading for the reader since there are no demonstration of reservoir computing in the work, even if the dynamics of the device can be potentially used for this application.

2) The mechanism of switching that give rise to device dynamics should be discussed.

3) A more detailed characterization of electrical properties of the device should be provided. The pristine state resistance of different electrode configuration should be reported. What is the device response to an applied voltage sweep? This can help in clarifying the switching mechanism

4) Authors discuss that the miniaturization of the device can potentially reduce the time required for device stimulation. What are the intrinsic limit of scaling these devices?

Author Response

Reviewer 2

1) The title could be misleading for the reader since there are no demonstration of reservoir computing in the work, even if the dynamics of the device can be potentially used for this application.

The title of the article was changed to avoid misleading the readers. We also made some corrections in the text to emphasize the potentiality of using such devices in reservoir computing systems instead of demonstrating state-of-the-art technology. We have also modified the Introduction section and added Conclusions section.

2) The mechanism of switching that give rise to device dynamics should be discussed.

Switching of PANI memristive devices is caused by oxidation and reduction of polyaniline that covers polyamide fibers. The redox processes are carried out with the help of ions dissolved in electrolyte gel. Transition to high-conductive (oxidation) and low-conductive (reduction) states occurs under different voltage biases (>0.5 V and <0.2 V, respectively) because of complex ionic dynamics. The device’s conductivity is quite stable within the range between these values, which is used to perform reading of the conductivity state. More detailed description of switching mechanism of PANI memristive devices with insights into ion dynamics and PANI redox processes can be found in the following articles:

Berzina, T., Erokhina, S., Camorani, P., Konovalov, O., Erokhin, V., Fontana, M.P., 2009. Electrochemical Control of the Conductivity in an Organic Memristor: A Time-Resolved X-ray Fluorescence Study of Ionic Drift as a Function of the Applied Voltage. ACS Appl. Mater. Interfaces 1, 2115–2118. https://doi.org/10.1021/am900464k

Lapkin, D.A., Korovin, A.N., Malakhov, S.N., Emelyanov, A.V., Demin, V.A., Erokhin, V.V., 2020. Optical Monitoring of the Resistive States of a Polyaniline‐Based Memristive Device. Adv. Electron. Mater. 6, 2000511. https://doi.org/10.1002/aelm.202000511

3) A more detailed characterization of electrical properties of the device should be provided. The pristine state resistance of different electrode configuration should be reported. What is the device response to an applied voltage sweep? This can help in clarifying the switching mechanism

Switching mechanism based on redox reactions in PANI. This fact is confirmed by oxidative and reductive peaks occurring in values of the current measured on the gate electrode, which is shown in (Lapkin et al., 2019). Switching processes for a single memristive device based on stochastic nonwoven material is shown in (Lapkin et al., 2019) and not provided here as the material has not changed. However, in the current work we show change in resistance of all the electrode configurations under +0.8 and -0.3 V voltage (Figure 4a). One can see the gradual change in resistance under voltage bias applied to the electrodes. In addition, the device demonstrates different output current response to application of voltage to different input terminals.

Lapkin, D.A., Malakhov, S.N., Demin, V.A., Chvalun, S.N., Feigin, L.A., 2019. Hybrid polyaniline/polyamide-6 fibers and nonwoven materials for assembling organic memristive elements. Synthetic Metals 254, 63–67. https://doi.org/10.1016/j.synthmet.2019.05.016

4) Authors discuss that the miniaturization of the device can potentially reduce the time required for device stimulation. What are the intrinsic limit of scaling these devices? 

From our view, the active area of the device (area between two planar electrodes) could be scaled down to several hundred of µm2 without obvious physical limitations. In one of our previous works we have shown a device based on a single fiber (Lapkin et al., 2019). However, single fiber is not enough to maintain the stochastic nature of the device since the diameter of the former is about 500 nm. In addition, the fabrication process is complicated by fragility of nonwoven material, which would be even worsened by small size. Thus significant miniaturization needs a much more sophisticated technological process (e.g. micromanipulator). Corresponding comments were added to the manuscript.

Lapkin, D.A., Malakhov, S.N., Demin, V.A., Chvalun, S.N., Feigin, L.A., 2019. Hybrid polyaniline/polyamide-6 fibers and nonwoven materials for assembling organic memristive elements. Synthetic Metals 254, 63–67. https://doi.org/10.1016/j.synthmet.2019.05.016

Round 2

Reviewer 2 Report

Authors have quite satisfactorely replied to reviewer questions. Thus, I suggest publication oin this journal.